# Causes of Mortality and Loss of Lumpfish *Cyclopterus lumpus*

Patrick Reynolds [1] , Albert Kjartan Dagbjartarson Imsland [2,3,*] and Lauris Boissonnot [4]

1 GIFAS AS, Gildeskål, 8140 Inndyr, Norway
2 Akvaplan-niva Iceland Office, Akralind 4, 201 Kopavogur, Iceland
3 Department of Biological Sciences, University of Bergen, High Technology Centre, 5020 Bergen, Norway
4 Aqua Kompetanse AS, 7770 Flatanger, Norway
* Correspondence: albert.imsland@uib.no

**Abstract:** Data from research and commercial use of lumpfish were collected from the research facilities of Gifas (Inndyr, Northern Norway, 67.0° N, 14.0° E). The data were sourced from 12 main lumpfish groups subdivided into 66 subgroups (N = 160,729) delivered to Gifas between 2013 and 2020 and used in cleaner-fish research in (a) land-based facilities, (b) small-scale, or (c) large-scale sea pens. The data were standardised and organised into three main headings. Firstly, background information included transfer time, point of origin, mean starting weight and population size on arrival. Other information included to which site the lumpfish were transferred, volume of cage/tank, whether in the presence of salmon or not, stocking density, days at each site and water quality parameters. Causes of mortality were recorded, when possible, for each group, along with calculated mortality rates, patterns, and analytical information, along with imaging where available. Results show that causes of mortality varied within and between research sites. For lumpfish in hatcheries as well as for those deployed at small-scale sea pens, the primary cause of mortality was identified as pathogenic, while for lumpfish deployed at large-scale sea pens, transporting, grading and mechanical delousing were the primary causes of mortality. The results indicate that more research is required to clarify best practices both in commercial hatcheries and salmon cages and further understanding on lumpfish biological requirements and stress physiology is necessary to develop better methods that safeguard lumpfish welfare and meet their needs.

**Keywords:** lumpfish; salmon farming; welfare; mechanical delousing; transport; pathogenic

## 1. Introduction

Lumpfish (*Cyclopterus lumpus*) have proven to be an effective lice eater at low sea temperatures [1–4]. However, high mortality and loss of cleaner-fish in salmon cages is one of the most serious problems the aquaculture industry in Norway faces at present. A study conducted by the Norwegian Food Safety Authority [5] revealed over 40% mortality of lumpfish deployed in Atlantic salmon net-pens in Norway. Results from the survey showed that farmers associate lumpfish mortalities with the occurrence of disease, but also handling and mechanical procedures, such as mechanical delousing. The study also found that lumpfish vaccination without anaesthesia is common practice, despite the lack of scientific studies showing any benefit of avoiding the use of anaesthesia during vaccination. This methodology can potentially contribute to events of extreme acute-stress responses that can often trigger other secondary issues [6]. Despite the common use of vaccination, the consensus among farmers was that existing vaccines were not effective and required further development. Findings based on additional assessments from the lumpfish and other cleaner-fish species used in Norwegian aquaculture are in accordance with the findings in [5], and [7,8] depict a grim scenario on lumpfish aquaculture, from hatcheries to Atlantic salmon farmers, where lumpfish welfare is questioned. The reports found a lack of standardized practices, inadequate feeding, lack of dedicated personnel to follow

up and constantly monitor health, and even lack of fundamental knowledge regarding species-specific needs [9].

The production methods for cultivation of lumpfish have been established [10] and are based on the technologies for rearing marine fin-fish such as sea bass and cod. Production of lumpfish has increased rapidly, and survival from hatch to stocking is high for a marine fin-fish species under hatchery conditions. There is the bonus that lumpfish grow quickly and can reach stocking size of 30 g within 6 to 8 months. However, there remain production, disease, vaccination, and quality issues. In many hatcheries, the suckers of the fish can be deformed and do not permit attachment to a substrate. These fish are unlikely to survive for long in sea pens, and even in the hatchery have to rest on the bottom of the tank, often on their side. These fish may be culled at an early stage. There may be genetic or nutritional causes for these abnormalities, but the condition needs to be fully investigated. In sum, there are various production-related issues that can contribute to the mortality of lumpfish in later stages and hence the need to look at underlying reasons for mortality and loss, as done in this study.

Worsening of lumpfish's welfare status has also been reported after transportation and deployment on sea pens, which can pose a significant stress. Conditions such as rapid cataract occurrence have also been demonstrated on deployed lumpfish, which is often associated with suboptimal feeds [11] or feeding protocols [12]

Although recent reports [5,7,8] highlighted the current grim situation, it was also pointed out that a significant part of the data were not included on the studies, as the reported data lacked coherency and method. This emphasizes the urgent need for standardization of methodology, such as in health monitoring, and the great potential for improving welfare by adopting strategies that safeguard lumpfish welfare. Continuous health and welfare monitoring are essential to help identify when and what procedures and operations are detrimental and thus adapt and improve practices [13,14]. To encourage the adoption of health status monitoring in a standardized, more comparable way, practical and user-friendly approaches are necessary. Recently, operational welfare indicators for lumpfish [15,16] have been published that can be used to define best-practice guides for better welfare and reduced mortality. For several years, Gifas in collaboration with Akvaplan-niva has conducted several large-scale and small-scale studies [1–4,10–12,15,17–23]. In these experiments, the focus has been on grazing effect, but information on mortality and loss has also been systematically gathered, which is presented in this study.

The main objective of the current study was to map the actual causes of mortality and loss of lumpfish fish, both in the hatchery phase and in the sea phase.

## 2. Materials and Methods

### 2.1. Lumpfish Research Sites and Data Mining

Data from research and commercial use of lumpfish were collected from all Gifas facilities where this species was utilised. The facilities were identified as (A) land-based (MH), (B) small-scale Langholmen (LH), and commercial farm sites (Røssøy, Leirvika Nord, and Halstenhamn (SS). Data were sourced from 12 main lumpfish groups subdivided into 66 subgroups (N = 160,729, Table 1) that were delivered to Gifas between 2013 and 2020. The number of lumpfish transferred to each facility was N = 400 for MH (4 subgroups), N = 1768 for LH (39 subgroups) and N = 158,564 for SS (23 subgroups).

The data were standardised and organised into three main headings. Firstly, background information included transfer time, point of origin, mean starting weight and population size on arrival. Other information included to which site the lumpfish were transferred, volume of cage/tank, whether in the presence of salmon or not, stocking density, days at each site, and water quality parameters. Causes of mortality were recorded, when possible, for each group, along with calculated mortality rates, patterns, and analytical information if available, along with imaging where available.

Table 1. Background information on lumpfish transferred to Gifas facilities in 2013–2020 and analysed for mortality and losses in this study.

| CODE | No. Code | | Origin | Transfer Month | Year | Transfer Method | Start Weigh (g) | Number of Cages | Diet Hatchery | Diet Pre-Transfer | Diet Post-Transfer | Population (N) | Site | Volume m³ | with Salmon | Salmon Weight | SD | Days at Site | Temp | DO |
|---|---|---|---|---|---|---|---|---|---|---|---|---|---|---|---|---|---|---|---|---|
| MH1 | MH1 | W | AC Lofoten | May | 2018 | road | 151.4 | 3 | ? | Biomar | Amber Nep | 90 | MH | 2.5 | N | | - | 68 | 7.3 | 92.5 |
| | MH2 | W | AC Lofoten | May | 2018 | road | 153.8 | 3 | | Biomar | Amber Nep | 90 | MH | 2.5 | N | | | 68 | 7.3 | 92.5 |
| MH2 | MH3 | W | ASG | Jan | 2020 | road | 32.8 | 2 | Otohime | Skretting | Skretting | 110 | MH | 2.5 | N | | | 61 | 6.5 | 90.0 |
| | MH4 | W | ASG | Jan | 2020 | road | 32.3 | 2 | Otohime | Skretting | Biomar | 110 | MH | 2.5 | N | | | 61 | 6.5 | 90.0 |
| LH1 | LH1 | W | AC Lofoten | Jan | 2015 | road | 22.6 | 2 | ? | ? | Amber Nep. | 34 | LH | 125 | Y | 538.2 | 10.0 | 159 | 6.4 | 91.2 |
| | LH2 | W | AC Lofoten | Jan | 2015 | road | 77.4 | 2 | ? | ? | Amber Nep. | 30 | LH | 125 | Y | 516.3 | 10.0 | 159 | 6.4 | 91.2 |
| | LH3 | W | AC Lofoten | Jan | 2015 | road | 113.5 | 2 | ? | ? | Amber Nep. | 30 | LH | 125 | Y | 546.0 | 10.0 | 159 | 6.4 | 91.2 |
| LH2 | LH4 | W | Trond | May | 2013 | road | 42.1 | 2 | ? | ? | Amber Nep. | 36 | LH | 125 | Y | 298.4 | 12.5 | 75 | 9.7 | 88.7 |
| | LH5 | W | Trond | May | 2013 | road | 80.1 | 2 | ? | ? | Amber Nep. | 36 | LH | 125 | Y | 297.8 | 12.5 | 75 | 9.7 | 88.7 |
| | LH6 | W | Trond | May | 2013 | road | 144.5 | 2 | ? | ? | Amber Nep. | 36 | LH | 125 | Y | 287.6 | 12.5 | 75 | 9.7 | 88.7 |
| LH3 | LH7 | W | Tromso | May | 2016 | road | 122.1 | 2 | ? | Amber Nep. | Amber Nep. | 60 | LH | 125 | Y | 653.7 | 10.0 | 62 | 9.1 | 93.5 |
| | LH8 | W | Tromso | May | 2016 | road | 113.2 | 2 | ? | Amber Nep. | Amber Nep. | 60 | LH | 125 | Y | 656.0 | 10.0 | 62 | 9.1 | 93.5 |
| LH4 | LH9 | W | Tromso | May | 2015 | road | 186.0 | 2 | Artemia/ Gemma | Gemma | Amber Nep. | 43 | LH | 125 | Y | 123.1 | 10.0 | 78 | 9.1 | 94.2 |
| | LH10 | W | Tromso | May | 2015 | road | 122.6 | 2 | Artemia/ Gemma | Gemma | Amber Nep. | 43 | LH | 125 | Y | 123.8 | 10.0 | 78 | 9.1 | 94.2 |
| | LH11 | W | Tromso | May | 2015 | road | 178.7 | 2 | Artemia/ Gemma | Gemma | Amber Nep. | 43 | LH | 125 | Y | 120.1 | 10.0 | 78 | 9.1 | 94.2 |
| | LH12 | W | Tromso | May | 2015 | road | 152.0 | 2 | Artemia/ Gemma | Gemma | Amber Nep. | 43 | LH | 125 | Y | 121.6 | 10.0 | 78 | 9.1 | 94.2 |
| | LH13 | W | Tromso | May | 2015 | road | 223.3 | 2 | Artemia/ Gemma | Gemma | Amber Nep. | 43 | LH | 125 | Y | 122.8 | 10.0 | 78 | 9.1 | 94.2 |
| | LH14 | W | Tromso | May | 2015 | road | 157.8 | 2 | Artemia/ Gemma | Gemma | Amber Nep. | 43 | LH | 125 | Y | 123.4 | 10.0 | 78 | 9.1 | 94.2 |
| | LH15 | W | Tromso | May | 2015 | road | 149.2 | 2 | Artemia/ Gemma | Gemma | Amber Nep. | 43 | LH | 125 | Y | 123.7 | 10.0 | 78 | 9.1 | 94.2 |
| | LH16 | W | Tromso | May | 2015 | road | 184.7 | 2 | Artemia/ Gemma | Gemma | Amber Nep. | 43 | LH | 125 | Y | 123.0 | 10.0 | 78 | 9.1 | 94.2 |
| | LH17 | W | Tromso | May | 2015 | road | 171.0 | 2 | Artemia/ Gemma | Gemma | Amber Nep. | 43 | LH | 125 | Y | 124.5 | 10.0 | 78 | 9.1 | 94.2 |
| LH5 | LH18 | W | Tromso | Sept | 2018 | road | 39.8 | 2 | Gemma | Gemma | feed blocks | 48 | LH | 125 | Y | 394.2 | 10.0 | 73 | 8.4 | 84.9 |
| | LH19 | W | Tromso | Sept | 2018 | road | 49.3 | 2 | Gemma | Gemma | feed blocks | 48 | LH | 125 | Y | 387.7 | 10.0 | 73 | 8.4 | 84.9 |

**Table 1.** *Cont.*

| CODE | No. Code | Origin | | Transfer Month | Transfer Year | Transfer Method | Start Weigh (g) | Number of Cages | Hatchery | Diet Pre-Transfer | Diet Post-Transfer | Population (N) | Site | Volume m³ | with Salmon | Salmon Weight | SD | Days at Site | Temp | DO |
|---|---|---|---|---|---|---|---|---|---|---|---|---|---|---|---|---|---|---|---|---|
| | LH20 | W/F | Tromso | Oct | 2019 | road | 55.3 | 2 | Gemma | Gemma | feed blocks | 48 | LH | 125 | Y | 642.4 | 10.0 | 69 | 7.9 | 89.4 |
| | LH21 | W/F | Tromso | Oct | 2019 | road | 39.2 | 2 | Gemma | Gemma | feed blocks | 48 | LH | 125 | Y | 624.6 | 10.0 | 69 | 7.9 | 89.4 |
| | LH22 | W/F | Tromso | Oct | 2019 | road | 56.9 | 2 | Gemma | Gemma | feed blocks | 48 | LH | 125 | Y | 627.7 | 10.0 | 69 | 7.9 | 89.4 |
| | LH23 | W/F | Tromso | Oct | 2019 | road | 69.3 | 2 | Gemma | Gemma | feed blocks | 48 | LH | 125 | Y | 611.3 | 10.0 | 69 | 7.9 | 89.4 |
| | LH24 | W/F | Tromso | Oct | 2019 | road | 56.4 | 2 | Gemma | Gemma | feed blocks | 48 | LH | 125 | Y | 618.2 | 10.0 | 69 | 7.9 | 89.4 |
| LH6 | LH25 | W/F | Tromso | Oct | 2019 | road | 44.4 | 2 | Gemma | Gemma | feed blocks | 48 | LH | 125 | Y | 614.6 | 10.0 | 69 | 7.9 | 89.4 |
| | LH26 | W/F | Tromso | Oct | 2019 | road | 67.1 | 2 | Gemma | Gemma | feed blocks | 48 | LH | 125 | Y | 615.9 | 10.0 | 69 | 7.9 | 89.4 |
| | LH27 | W/F | Tromso | Oct | 2019 | road | 55.4 | 2 | Gemma | Gemma | feed blocks | 48 | LH | 125 | Y | 626.3 | 10.0 | 69 | 7.9 | 89.4 |
| | LH28 | W/F | Tromso | Oct | 2019 | road | 56.8 | 2 | Gemma | Gemma | feed blocks | 48 | LH | 125 | Y | 618.6 | 10.0 | 69 | 7.9 | 89.4 |
| | LH29 | W/F | Tromso | Oct | 2019 | road | 46.9 | 2 | Gemma | Gemma | feed blocks | 48 | LH | 125 | Y | 614.3 | 10.0 | 69 | 7.9 | 89.4 |
| | LH30 | W/F | Tromso | Jul | 2020 | road | 35.4 | 2 | Gemma | Gemma | feed blocks | 48 | LH | 125 | Y | 291.3 | 10.0 | 77 | 12.9 | 89.5 |
| | LH31 | W/F | Tromso | Jul | 2020 | road | 40.5 | 2 | Gemma | Gemma | feed blocks | 48 | LH | 125 | Y | 309.4 | 10.0 | 77 | 12.9 | 89.5 |
| | LH32 | W/F | Tromso | Jul | 2020 | road | 39.7 | 2 | Gemma | Gemma | feed blocks | 48 | LH | 125 | Y | 284.4 | 10.0 | 77 | 12.9 | 89.5 |
| | LH33 | W/F | Tromso | Jul | 2020 | road | 47.7 | 2 | Gemma | Gemma | feed blocks | 48 | LH | 125 | Y | 278.4 | 10.0 | 77 | 12.9 | 89.5 |
| | LH34 | W/F | Tromso | Jul | 2020 | road | 38.0 | 2 | Gemma | Gemma | feed blocks | 48 | LH | 125 | Y | 286.1 | 10.0 | 77 | 12.9 | 89.5 |
| LH7 | LH35 | W/F | Tromso | Jul | 2020 | road | 42.8 | 2 | Gemma | Gemma | feed blocks | 48 | LH | 125 | Y | 259.6 | 10.0 | 77 | 12.9 | 89.5 |
| | LH36 | W/F | Tromso | Jul | 2020 | road | 48.9 | 2 | Gemma | Gemma | feed blocks | 48 | LH | 125 | Y | 271.6 | 10.0 | 77 | 12.9 | 89.5 |
| | LH37 | W/F | Tromso | Jul | 2020 | road | 47.3 | 2 | Gemma | Gemma | feed blocks | 48 | LH | 125 | Y | 267.6 | 10.0 | 77 | 12.9 | 89.5 |
| | LH38 | W/F | Tromso | Jul | 2020 | road | 40.3 | 2 | Gemma | Gemma | feed blocks | 48 | LH | 125 | Y | 263.6 | 10.0 | 77 | 12.9 | 89.5 |
| | LH39 | W/F | Tromso | Jul | 2020 | road | 39.6 | 2 | Gemma | Gemma | feed blocks | 48 | LH | 125 | Y | 259.6 | 10.0 | 77 | 12.9 | 89.5 |
| | SS1 | W | ASG | Feb | 2019 | WB | 70.1 | 1 | ? | Skretting | feed blocks | 14074 | RØSS | 140M PC | Y | 533.4 | 8.0 | 10 | 4.9 | 93.2 |
| SS8 | SS2 | W | ASG | Feb | 2019 | WB | 69.5 | 1 | ? | Skretting | Skretting | 13977 | RØSS | 140M PC | Y | 499.5 | 8.0 | 10 | 4.9 | 93.2 |
| | SS3 | W | ASG | Feb | 2019 | WB | 72.8 | 1 | ? | Skretting | feed blocks | 13985 | RØSS | 140M PC | Y | 556.2 | 8.0 | 10 | 4.9 | 93.2 |
| | SS4 | W | ASG | Feb | 2019 | WB | 73.4 | 1 | ? | Skretting | Skretting | 11842 | RØSS | 140M PC | Y | 453.8 | 8.0 | 10 | 4.9 | 93.2 |
| | SS5 | W | ASG | Sep | 2018 | WB | 52.2 | 1 | Gemma | Skretting | feed blocks | 3800 | LEIRV | 90M PC | Y | 1966.7 | 8.0 | 181 | 9.1 | 93.7 |
| | SS6 | W | ASG | Sep | 2018 | WB | 55.8 | 1 | Gemma | Skretting | feed blocks | 3800 | LEIRV | 90M PC | Y | 1941.2 | 8.0 | 181 | 9.1 | 93.7 |
| | SS7 | W | ASG | Sep | 2018 | WB | 56.1 | 1 | Gemma | Skretting | feed blocks | 3800 | LEIRV | 90M PC | Y | 1726.5 | 8.0 | 181 | 9.1 | 93.7 |
| | SS8 | W | ASG | Sep | 2018 | WB | 49.5 | 1 | Gemma | Skretting | feed blocks | 3800 | LEIRV | 90M PC | Y | 1873.5 | 8.0 | 181 | 9.1 | 93.7 |
| | SS9 | W | ASG | Sep | 2018 | WB | 52.6 | 1 | Gemma | Skretting | Skretting | 3800 | LEIRV | 90M PC | Y | 1487.5 | 8.0 | 181 | 9.1 | 93.7 |
| | SS10 | W | ASG | Sep | 2018 | WB | 49.9 | 1 | Gemma | Skretting | Skretting | 3800 | LEIRV | 90M PC | Y | 1938.4 | 8.0 | 181 | 9.1 | 93.7 |
| SS9 | SS11 | W | ASG | Sep | 2018 | WB | 58.5 | 1 | Gemma | Skretting | Skretting | 3800 | LEIRV | 90M PC | Y | 1445.9 | 8.0 | 181 | 9.1 | 93.7 |
| | SS12 | W | ASG | Sep | 2018 | WB | 53.5 | 1 | Gemma | Skretting | Skretting | 3800 | LEIRV | 90M PC | Y | 1771.0 | 8.0 | 181 | 9.1 | 93.7 |
| | SS13 | W | ASG | Sep | 2018 | WB | 51.4 | 1 | Gemma | Skretting | Skretting | 3800 | LEIRV | 90M PC | Y | 1706.4 | 8.0 | 181 | 9.1 | 93.7 |
| | SS14 | W | ASG | Sep | 2018 | WB | 52.2 | 1 | Gemma | Skretting | Skretting | 3800 | LEIRV | 90M PC | Y | 1660.7 | 8.0 | 181 | 9.1 | 93.7 |
| | SS15 | W | ASG | Sep | 2018 | WB | 54.7 | 1 | Gemma | Skretting | Skretting | 3800 | LEIRV | 90M PC | Y | 1542.4 | 8.0 | 181 | 9.1 | 93.7 |
| | SS16 | W | ASG | Sep | 2018 | WB | 60.1 | 1 | Gemma | Skretting | Skretting | 3800 | LEIRV | 90M PC | Y | 1665.5 | 8.0 | 181 | 9.1 | 93.7 |
| | SS17 | W | ASG | Sep | 2018 | WB | 49.5 | 1 | Gemma | Skretting | Skretting | 3800 | LEIRV | 90M PC | Y | 1437.3 | 8.0 | 181 | 9.1 | 93.7 |

**Table 1.** *Cont.*

| CODE | No. Code | Origin | | Transfer Month | Year | Transfer Method | Start Weigh (g) | Number of Cages | Hatchery | Diet Pre- Transfer | Post-Transfer | Populat-ion (N) | Site | Volume m³ | with Salmon | Salmon Weight | SD | Days at Site | Water Quality Temp | DO |
|------|------|---|---|------|------|------|------|------|------|------|------|------|------|------|------|------|------|------|------|------|
| | SS18 | W | ASG | Aug | 2019 | WB | 52.5 | 1 | Gemma | Skretting | feed blocks | 7965 | HALST | 140M PC | Y | 808.3 | 8.0 | 272 | 7.4 | 92.8 |
| | SS19 | W | ASG | Aug | 2019 | WB | 50.1 | 1 | Gemma | Skretting | feed blocks | 10775 | HALST | 140M PC | Y | 755.5 | 8.0 | 272 | 7.4 | 92.8 |
| | SS20 | W | ASG | Aug | 2019 | WB | 56.8 | 1 | Gemma | Skretting | feed blocks | 7876 | HALST | 140M PC | Y | 268.7 | 8.0 | 272 | 7.4 | 92.8 |
| SS10 | SS21 | W | ASG | Aug | 2019 | WB | 52.7 | 1 | Gemma | Skretting | Skretting | 7766 | HALST | 140M PC | Y | 564.9 | 8.0 | 272 | 7.4 | 92.8 |
| | SS22 | W | ASG | Aug | 2019 | WB | 53.6 | 1 | Gemma | Skretting | Skretting | 7823 | HALST | 140M PC | Y | 696.9 | 8.0 | 272 | 7.4 | 92.8 |
| | SS23 | W | ASG | Aug | 2019 | WB | 55.7 | 1 | Gemma | Skretting | Skretting | 13081 | HALST | 140M PC | Y | 168.1 | 8.0 | 272 | 7.4 | 92.8 |

Abbreviations: Origin: W = wild; W/F = first generation domesticated; Transport: Road = transport by truck on land; WB = transport by well-boat; Site: MH = land based; LH = small sea pens; RØSS, LEIRV, HALST = large-scale sea pens.

## 2.2. Background Information

Of the 66 subgroups analysed in the study, 46 were derived from wild broodstock and 20 were derived from farmed and wild broodstock that were part of an ongoing breeding programme on lumpfish. All background information recorded for each group is summarised in Table 1. Lumpfish groups were transferred from four main suppliers between 2013 and 2020. All groups transferred to land-based and small-scale facilities were transferred by road whilst all lumpfish transferred to Gifas commercial farm sites were transferred by well-boat. All groups transferred to sea were stocked with Atlantic salmon.

All four groups of lumpfish were transferred to MH (land-based) during January and February whilst for lumpfish groups transferred to LH (small-scale), three were transferred in January and 14 and 10 groups were transferred during May and July, respectively. Of the remaining groups, 2 were transferred during September and 10 during October. The majority of lumpfish transferred to SS (commercial cage sites) occurred during the month of September (13) whilst 4 and 6 groups were transferred during February and August, respectively.

Of the four subgroups transferred to MH, mean weights ranged between 32.3 g and 153.8 g. Lumpfish subgroups transferred to LH ranged between 35.5 g and 223.3 g whilst subgroups transferred to SS ranged between 49.5 g and 73.4 g.

The two main lumpfish groups transferred to MH remained on site for 68 and 61 days respectively, while the seven main groups transferred to LH remained on site between 62 and 159 days. The three main groups transferred to SS were maintained on site between 63 and 272 days.

## 2.3. Health Assessment of Lumpfish

Assessment of the health status of 37 of the lumpfish subgroups was undertaken during routine sampling points using the Gifas Lumpfish Health Scoring System (LHSS, Table 2). The health scoring system focused on morphological health indicators for lumpfish. It is divided in several categories, where each is evaluated and scored accordingly by the user (Table 2). The assessment of health status was non-destructive, and lumpfish were returned to their specific cage after assessment. Each category has a specific "weight" in the final consideration of the overall health score. The specific attribution of an added "weight" for each category was decided and adjusted after appropriate testing with historical health data sets. The weighting criteria also consider some deteriorating conditions more severe than others and accordingly more weighting is applied in these instances. The input of score values in each category was calculated, giving a weighted health score for each fish.

The average group health score was calculated and an action highlighted based on the score (Table 2). If scores were between 0 and 3, health status was deemed satisfactory, and no action was required. A score of between 3 and 5 indicated health status had deteriorated and action was required. A score of over 5 indicated extensive health deterioration and immediate action required to alleviate suffering. In addition to the external condition of the lumpfish, evidence of any continual individual loss of growth and/or mortality rates was assessed. Condition was assessed using regression analysis for estimation of length and weight parameters. The relationship between weight (W) and length (L) in fishes has the form:

$$W = aL^b$$

The shape parameter b was calculated using historical weight, length, width, and height data from lumpfish (N = 3657). The results were used as part of the welfare scoring system utilised in this study.

All lumpfish deployed at Gifas facilities are routinely assessed for health status as mentioned above, which is based on external morphological examination. The system allows for further assessment if lumpfish appear distressed and/or mortalities are recorded. Fish are sampled for analysis of internal parameters that may be directly related to the cause of death/distress. These parameters include such assessments as liver colour, ascites, and abnormalities to internal organs, including signs of pathogenic presence. A range of

tissue types are sampled and extensively analysed using the appropriate methods, which include PCR, histology, bacteriology, and blood sampling.

**Table 2.** Lumpfish Health Scoring System (LHSS) utilized in this study.

| Individual Scoring Guide | | | | | | |
|---|---|---|---|---|---|---|
| **Fin Condition** | Erosion/Splitting | No visible damage | Less than 25% of the fin eroded—minor splitting | Between 25 and 50% of erosion | More than 50% erosion | |
| **Skin** | Body lesions/Wounds/ Inflammation | Intact | Minor injury/light inflammation | Increased localized damage | Open wounds/haemorrhaging | |
| **Malformations** | Suction disc Spine | Normal | Functional—light deformity | Functional—Obvious malformation | Non-functional—Severe deformity | |
| **Cataracts** | Size | No cataract | 0–10% of the eye | 10–40% of the eye | 40–70% of the eye | Over 70% of the eye |
| | Opacity | | Translucent | Opaque- Whitish crystalline | Totally opaque, loss of translucency | |
| **Eye damage** | Lesions/Ulcers/ Swelling | No visible damage | 0–25% of the eye | 25–50% of the eye | 50–75% of the eye | Over 75% of the eye |
| **Fitness** | Nutritional fitness | Optimal predicted weight | 15% to 25% below or 25% to 35% above optimal predicted weight | 25% to 50% below or 35% above optimal predicted weight | More than 50% below optimal predicted weight | |
| | | 0 | 1 | 2 | 3 | 4 |

| Group Evaluation and Action Guide | | |
|---|---|---|
| **Health score** | **Evaluation** | **Action** |
| 0–3 | No to minimal health deterioration | No action required |
| 3–5 | Signs of health deterioration | Measurements to improve health. Potential sampling to determine causes of health deterioration. |
| +5 | Compromised welfare | Consider removal of fish. Approved veterinary should be contacted. Potential additional samples to determine causes of such health deterioration. |

### 2.4. Statistical Methods

All statistical analyses were conducted using Statistica™ 12.0 software. Possible differences in mortality data were tested for each group with one-way analysis of variance (ANOVA). Significant differences revealed in ANOVA were followed by Tukey's multiple range tests to determine differences among experimental groups. A significance level ($\alpha$) of 0.05 was used if not stated otherwise. The cumulative percentage mortality data sets were subjected to linear regression analysis ($\alpha$ = 0.05). In all regressions, cumulative mortality data were the independent variable (X-axis) and respective response criteria was the dependent (Y) variable. Linear regression was also tested for significant deviation of the slope from zero ($p$-value, <0.05).

### 3. Results

#### 3.1. Causes of Mortality

There were nine known primary causes of mortality identified for all groups from the three locations, and mortalities with no proven cause were classified as unknown (Figure 1). The most frequent causes were handling/grading (21.2%), mechanical delousing (19.7%), and bacterial infections (16.7%), and 19.7% had no known cause. Mortality during transfer accounted for 6.1%, while 4.5% was linked to parasitic agents. Both viral infections and severe cataracts accounted for 3% of groups, and 1.5% was linked to dietary effects.

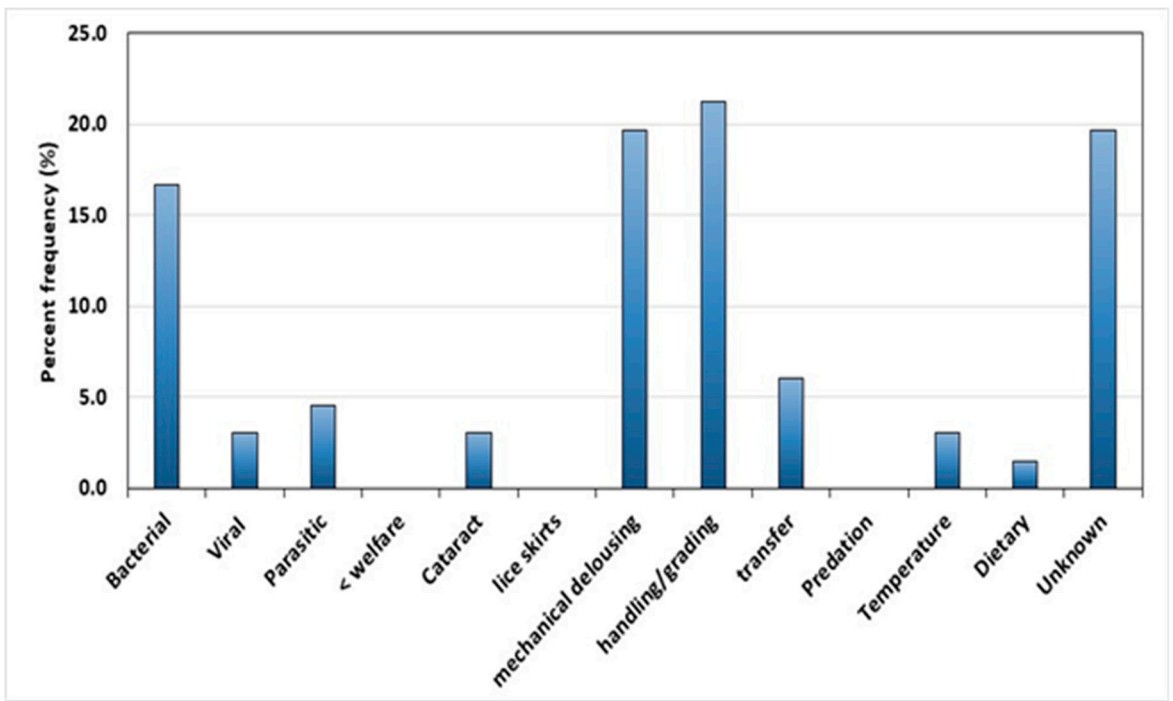

**Figure 1.** Percentage of primary causes of mortality identified for all lumpfish groups.

For each facility (Table 1), of the four groups transferred to MH, two (50%) had occurrences of mortality associated with bacterial infection whilst one group (25%) had mortalities linked to dietary effects. Of the 39 groups exhibiting mortality transferred to LH, bacterial agents were identified as the primary cause for nine of the groups (23.1%) whilst viral agents were responsible for mortalities in two of the groups (5.1%). Gill parasites were the primary cause of mortality in three groups (7.7%) whilst severe incidence of cataracts was accountable in two of the groups. Handling was identified as the primary cause of mortality for eight of the groups whilst temperature gradients were identified as the main causal agent for two groups. Thirteen groups (33.3%) had mortalities where no primary agent could be identified (Table 3). Of the 23 groups of lumpfish transferred to SS, 13 (56.5%) had mortalities where mechanical delousing was identified as the most likely primary cause, whilst grading and splitting of Atlantic salmon was the primary cause of mortality in six (26.1%) groups. Transporting lumpfish from the hatchery to sea cages accounted for mortality in four groups (Table 3).

**Table 3.** Number and percentage of primary causes of mortality recorded at each location.

| Causes | Land-Based (MH) | | Small-Scale (LH) | | Commercial (SS) | |
|---|---|---|---|---|---|---|
| | Number | Percentage | Number | Percentage | Number | Percentage |
| Bacterial. | 2 | 50.0 | 9 | 23.1 | - | - |
| Viral | - | - | 2 | 5.1 | - | - |
| Parasite | - | - | 3 | 7.7 | - | - |
| <Welfare | - | - | - | - | - | - |
| Cataract | - | - | 2 | 5.1 | - | - |
| Mechanical delousing | - | - | - | - | 13 | 56.5 |
| Grading/handling | - | - | 8 | 20.5 | 6 | 26.1 |
| Transporting | - | - | - | - | 4 | 17.4 |
| Predation | - | - | - | - | - | - |
| Temperature | - | - | 2 | 5.1 | - | - |
| Dietary | 1 | 25.0 | - | - | - | - |
| Unknown | - | - | 13 | 33.3 | - | - |

There were 11 known primary and secondary causes of morality identified for all groups from the three locations, and mortality with no proven cause was classified as unknown (Table 4). There were no secondary causes of mortality associated with any of the four groups transferred to MH (Table 4), while secondary causes of mortality of lumpfish transferred to LH were associated with handling, transporting, and unknown factors (Table 4). Secondary causes of mortalities of lumpfish transferred to SS were identified as being associated with bacterial infections, reduced welfare, grading, predation by birds, and unknown factors (Table 4).

**Table 4.** Number and percentage of primary and secondary causes of mortality recorded at each location. Number and percentage of secondary causes are in red with parenthesis.

| Causes | Land-Based (MH) | | Small-Scale (LH) | | Commercial (SS) | |
|---|---|---|---|---|---|---|
| | Number | Percentage | Number | Percentage | Number | Percentage |
| Bacterial. | 2 | 50.0 | 9 | 23.1 | 0 (4) | 0 (17.4) |
| Viral | - | - | 2 | 5.1 | - | - |
| Parasite | - | - | 3 | 7.7 | - | - |
| <Welfare | - | - | - | - | 0 (19) | 0 (82.6) |
| Cataract | - | - | 2 | 5.1 | - | - |
| Mechanical delousing | - | - | - | - | 13 | 56.5 |
| Grading/handling | - | - | 8 (1) | 20.5 (2.6) | 6 (6) | 26.1 (26.1) |
| Transporting | - | - | 0 (2) | 0 (5.1) | 4 | 17.4 |
| Predation | - | - | - | - | 0 (19) | 0 (82.6) |
| Temperature | - | - | 2 | 5.1 | - | - |
| Dietary | 1 | 25.0 | - | - | - | - |
| Unknown | - | - | 13 (1) | 33.3 (2.6) | 0 (16) | 0 (69.6) |

### 3.2. Mortality Patterns

3.2.1. Commercial Sites

The mean percentage weekly mortality of lumpfish deployed at Gifas commercial site Røssøy can be seen in Figure 2. The mean percentage mortality ($\pm$SD) at week one for all four polar circle cages stocked with lumpfish was calculated as 19.8% $\pm$ 4.3%. Mean weekly mortality decreased sharply to 1.8% $\pm$ 0.6% at week two onwards until week nine, when mean weekly mortality increased to 4.8% $\pm$ 1.7%. Mean weekly temperature slightly increased from 4.7 °C in week one to 5.9 °C in week nine.

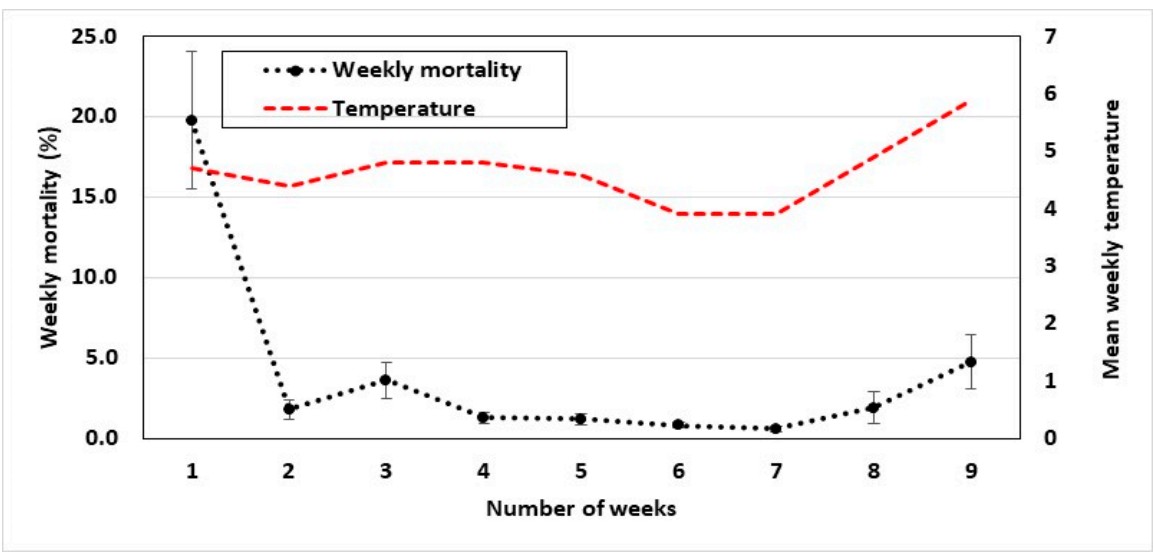

**Figure 2.** Percentage weekly mortality of lumpfish deployed at Gifas commercial site Røssøy. Red line indicates mean weekly temperature at the site.

The mean percentage weekly mortality of lumpfish deployed at Gifas commercial site Leirvika can be seen in Figure 3. The mean percentage mortality (±SD) remained under 1% for all groups during the 23-week deployment. Mean weekly mortality gradually increased through time, with the highest mean mortality of 0.8% ± 0.1% recorded during weeks 21 and 23. Temperature decreased from a mean of 13.1 °C in week one to 6.2 °C in week 23.

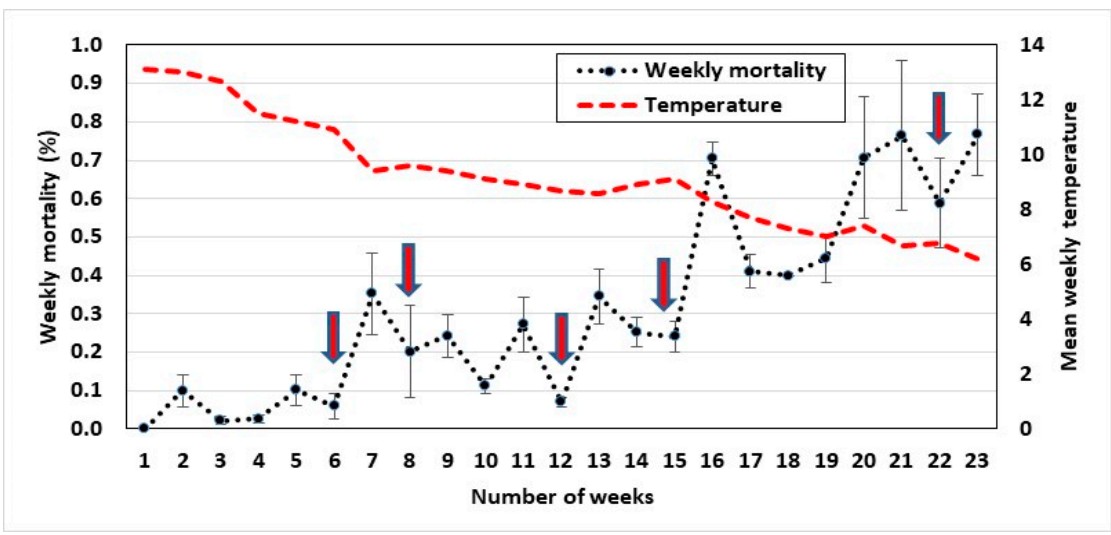

**Figure 3.** Percentage weekly mortality of lumpfish deployed at Gifas commercial site Leirvika. Red line indicates mean weekly temperature at the site. Arrows indicate when mechanical delousing was used in all cages.

The mean percentage weekly mortality of lumpfish deployed at Gifas commercial site Hallsteinhamn can be seen in Figure 4 The mean percentage mortality (±SD) remained low throughout for all subgroups, with weekly mean mortality rates ranging between 0 and 0.3% at week 1 and 1.2% recorded at week 15. Mean weekly temperature decreased from 8.0 °C in week 1 to 4.2 °C in week 37.

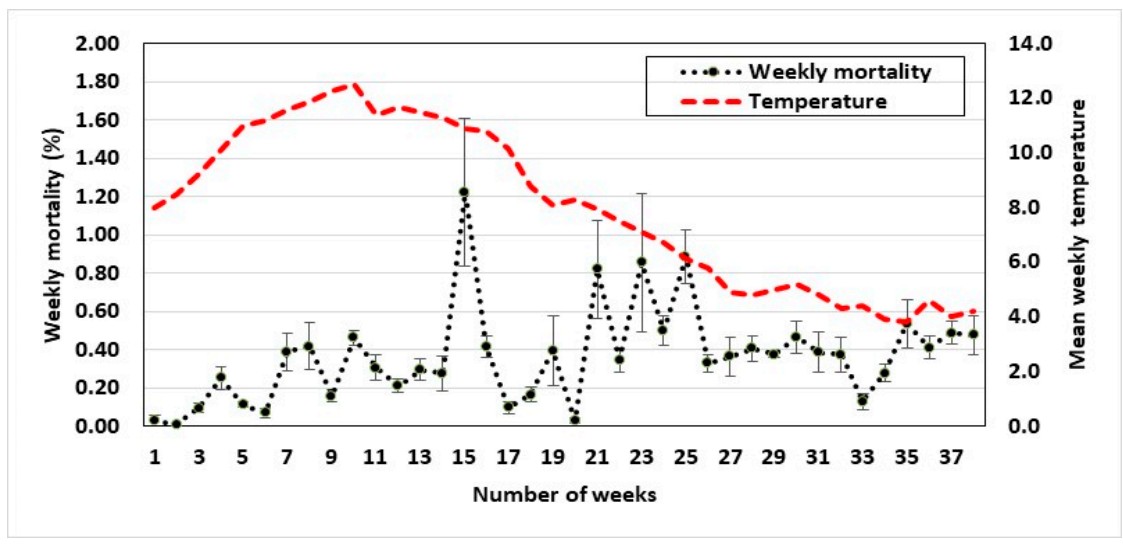

**Figure 4.** Percentage weekly mortality of lumpfish deployed at Gifas commercial site Hallsteinhamn. Red line indicates mean weekly temperature at the site.

### 3.2.2. Land-Based and Small-Scale Facilities

The mean percentage daily mortality of two lumpfish subgroups deployed at the land-based facility (MH) can be seen in Figure 5. The mean percentage mortality (±SD) for

subgroup K6 remained low throughout until day 41, when a daily percentage mortality of 4.3% was recorded. For subgroup K5, mortality rates remained low until day 34, when daily percentage mortality of 18.6% was recorded. This increased to 35.7% at day 38. Mortality rates fluctuated after this time point, but remained higher than subgroup K6. Mean daily temperature fluctuated between 7.0 °C at day 1 to 7.4 °C at day 48.

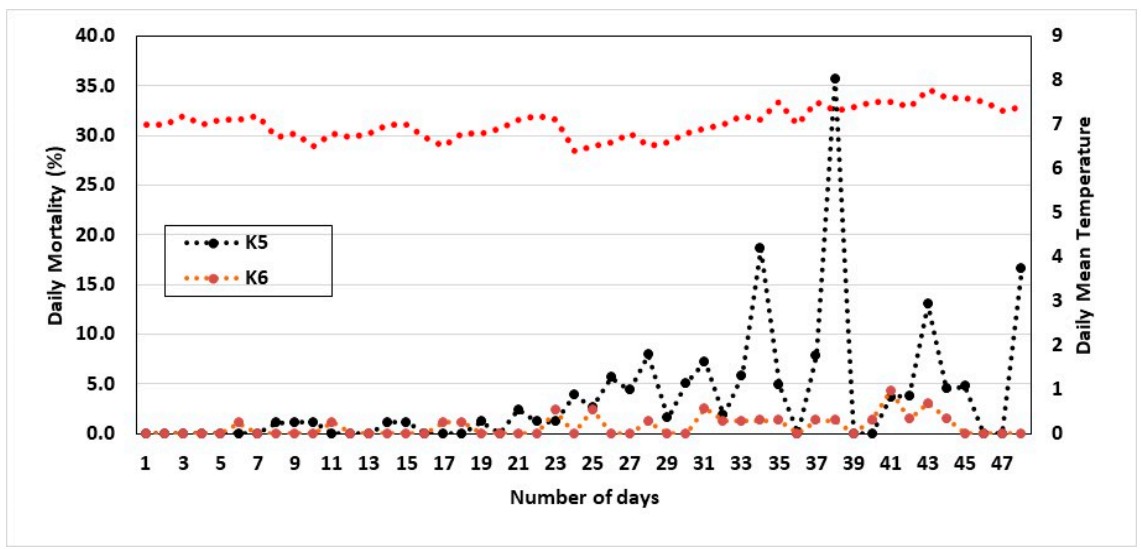

**Figure 5.** Percentage weekly mortality of lumpfish deployed at land-based facility (MH).

The mean percentage daily mortality of three lumpfish subgroups deployed at the small-scale facility (LH) can be seen in Figure 6. The mean daily percentage mortality (±SD) of the large lumpfish (444.6 g mean starting weight) remained low throughout, with the highest rate of 2.9% occurring at day 73, while daily percentage mortality of the medium lumpfish (80.1 g) peaked at day 62, when 10% mortality was recorded. Daily mortality decreased to low levels after this peak; however, it remained at levels of between 3.4% and 4.0% on several occasions. Daily mortality rates of the small lumpfish (42.1 g) remained at nil until day 56, when 2.8% was recorded. Daily mortality spiked on subsequent days, reaching a high of 12.5% at day 75. Daily mean temperature increased from 4.5 °C at day 1 to 12.0 °C at day 73.

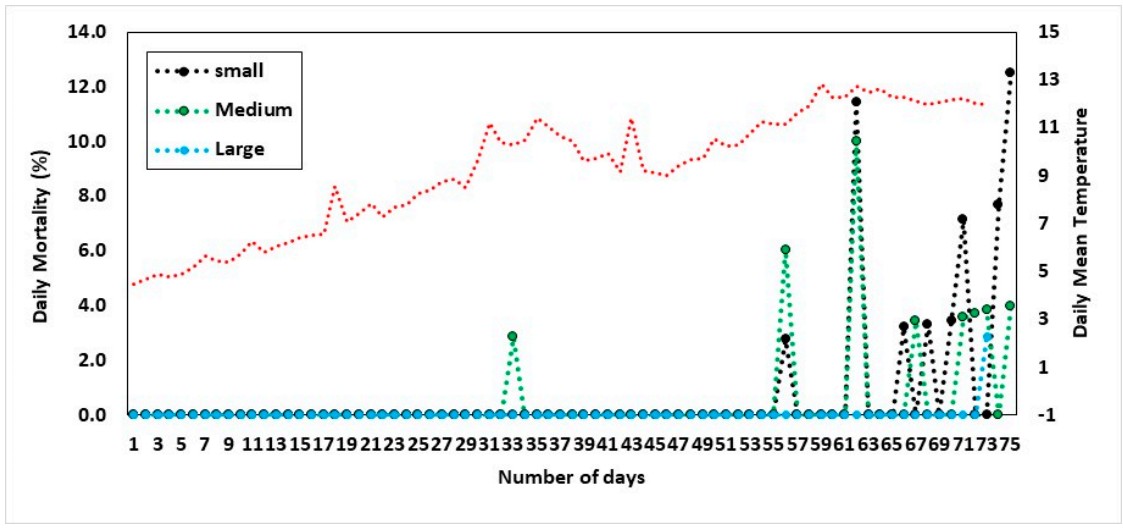

**Figure 6.** Percentage weekly mortality of three subgroups of lumpfish deployed at small-scale facility (LH). Subgroup small (42.1 g), subgroup medium (80.1 g), and subgroup large (444.6 g).

### 3.3. Cumulative Mortality

Cumulative mortality plotted against the primary causal agent of mortality for lumpfish groups deployed at commercial sites (SS) can be seen in Figure 7. The four groups with the primary cause of mortality identified as transporting had the highest percentage cumulative mortality compared to the other groups deployed at SS, while groups where the primary cause of mortality was identified as being either grading and/or mechanical delousing had similar cumulative mortality rates, ranging between 14.3% and 3.7%.

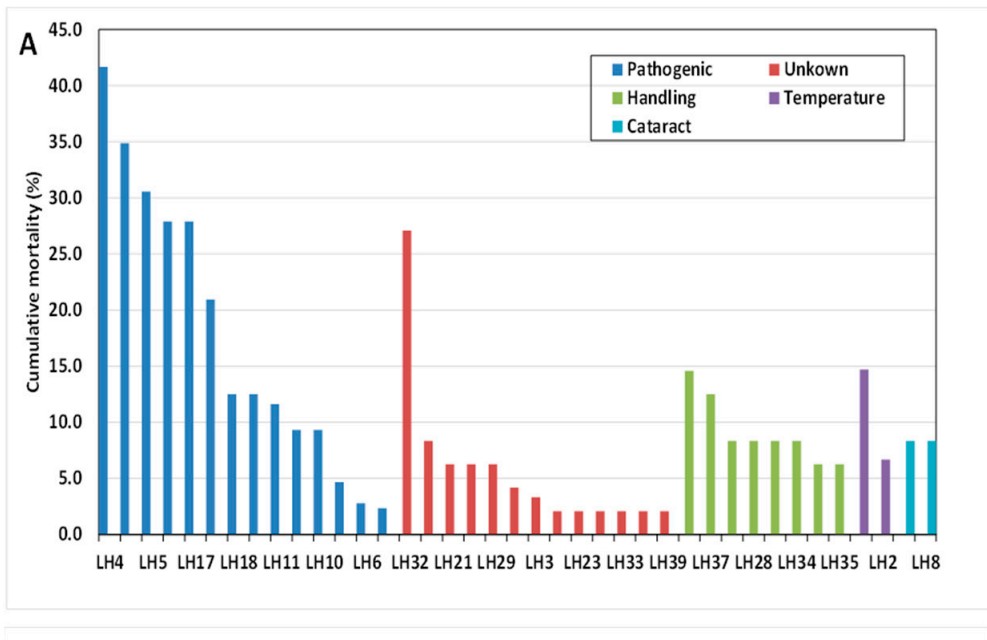

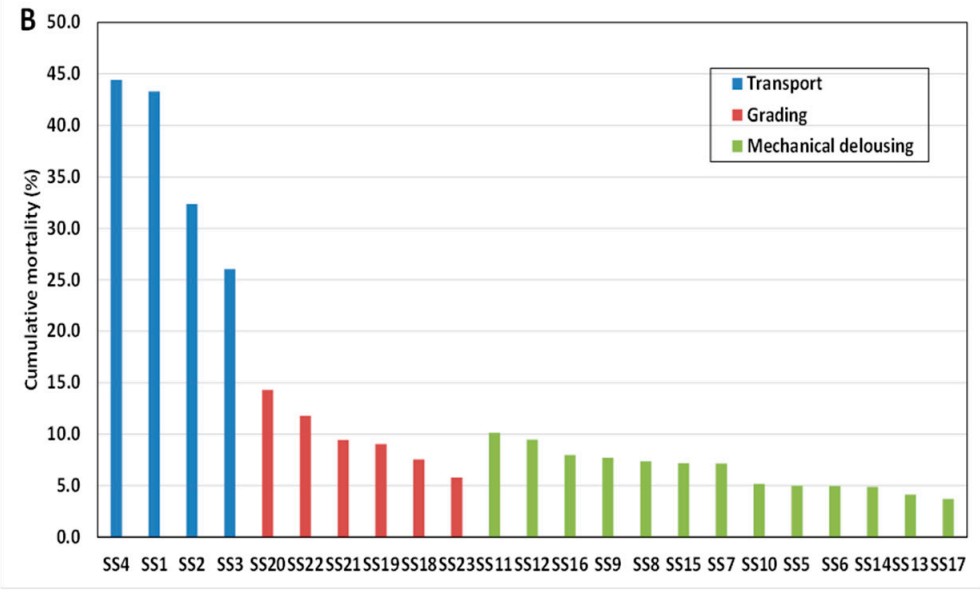

**Figure 7.** Percentage cumulative mortality for lumpfish groups plotted against primary cause of mortality for different groups (abbreviated on X axis) of lumpfish deployed at (**A**) small-scale and (**B**) commercial sites. See Table 1 for details of groups.

## 4. Discussion

Results from this study identified 10 primary causes of mortality in all the lumpfish groups deployed. Of these, handling/grading and mechanical delousing accounted for 21% and 20%, respectively, whilst bacterial infections accounted for 17% of all primary causes. Other primary causes of mortality included viral (3%), parasitic (5%), severe

cataract prevalence (3%), mortality during transfer to sea cages (6%), temperature gradients (3%), and dietary (2%). While it would be reasonable to expect causes of mortality in lumpfish populations to vary both from hatchery to hatchery but also geographically, there are evidently many reasons for mortality. Interestingly, mortality without a specific diagnosis accounted for 20% for the groups in the present study, but for 56% of the groups investigated by the Norwegian Veterinary Institute in 2012-14 [24]. Given that all causes of mortality were investigated for the primary causal agent in the present study, then perhaps the analytical diagnostic tools employed to determine causal factors need to be more extensively researched. Further understanding on lumpfish biological requirements and stress physiology is necessary to develop better methods that safeguard lumpfish well-being and meet their needs. Continuous health and welfare monitoring are essential to help identify when and what procedures and operations are detrimental and thus adapt and improve practices [13,14]. To encourage the adoption of health status monitorization in a standardized, more comparable way, practical and user-friendly approaches are necessary.

Stress and handling during transfer and stocking, together with the inevitable disease challenges in the pen environment, make the first few weeks at sea a critical time and acute mortalities often occur [25]. Stress and damage through events such as capture, handling, transport, or grading increase disease susceptibility and high mortality can follow these events. Lumpfish, despite their robust appearance, are very susceptible to skin damage and incidence is increased by their habit of adhering to substrates from which they are at times forcefully removed. A wide range of factors influence the welfare of cleaner-fish, and it is often the sites that put most effort into welfare that have the fewest disease problems (P. Reynolds, pers. comm.).

Health management of cleaner-fish in salmon pens can be challenging [25]. Often, primary factors that affect welfare and survival can facilitate secondary factors, thus further exacerbating an already worsening situation. In addition, often more than one pathogen is isolated from diseased fish, making it difficult to verify the primary cause of disease and death. The cause of death may be multifactorial and influenced by poor nutritional input and/or inadequate feeding strategies. Extreme environmental conditions may also be a factor, and studies have shown that both low temperatures (<4°C) [26] and high temperatures (18 °C) [27] can increase mortality. High mortality can also occur after transfer to commercial cages due to poor handling and has been linked to chemical and mechanical delousing practices, e.g., [9,28]. Further, mortalities have been attributed to net cleaning, bath treatments, or other operations.

All fish groups transferred to Gifas had evidence of cataracts detected either prior to or immediately after transfer. For fish groups screened for eye health and transferred to either small-scale facility, cataract prevalence varied between 0 and 29.2% upon transfer, and for commercial cages, prevalence varied between 12.1% and 79.6%. There was a tendency for prevalence to increase in all groups, with some groups approaching 90% prevalence and the severity of cataracts increasing with time. It is known that cataracts can affect how efficiently fish catch natural feed, such as in Arctic char (*Salvelinus alpinus*), where fish with no cataracts caught zooplankton more effectively than fish with cataracts [29]. However, a previous study [12] showed that a low degree of cataracts in lumpfish did not affect their ability to detect and consume sea lice or their overall feed intake and growth negatively. Regardless of some fish being able to find food items and maintain growth, the high proportion of fish in this study observed to be losing weight indicates that fish with moderate to severe cataracts cannot maintain their growth potential and ultimately health may be impaired. Mortality is affected by cataracts to the extent that they affect feed intake, growth and weakened immunity and robustness of the fish [30]. The average cataract index in the present study was generally low for most groups at the start of the deployment period, but generally increased with time. Previous studies have recorded a low cataract index in lumpfish [31], that there was no systematic relationship between growth and cataract index that could indicate that cataracts had an impact on the growth rate, or high SGR increased the risk of developing cataracts.

Previous studies have shown that low water temperatures may be a causal factor in reduced welfare and lower grazing performance of lumpfish (Gifas, unpublished data). Small lumpfish may be directly challenged when transferred to an open net-pen environment in winter, when most commercial salmon farms are routinely setting out juvenile lumpfish at an average weight of 30 g in commercial cages. Wild juvenile lumpfish spend their early stages in the physically challenging intertidal zone, where they are reported to grow rapidly before migrating to colder feeding grounds [32], where they may better exploit the differences in temperature. Studies of wild larval and juvenile lumpfish growth patterns show a rapid increase in growth rate from mid-July to August before decreasing in August–September [33,34].

In the present study, cumulative mortality rates were similar for groups deployed at small- and large-scale research facilities. The similarities in cumulative mortality between both sea-based facilities indicate that the different stressors that were identified as prime causes at each site were challenging at the same levels. For lumpfish deployed at LH, the primary cause of mortality identified as pathogenic generally had the highest percentage cumulative mortality compared to the other groups deployed, while for lumpfish deployed at large-scale research facilities, transporting, grading, and mechanical delousing had the highest percentage cumulative mortality. Thus, different stressors were evident dependent on site conditions, with lumpfish deployed in commercial cages more likely to be exposed to more mechanical treatments, such as transferring, grading, splitting of cages, and mechanical delousing, while for small-scale cages, repeated handling due to intensive sampling regimes may be the trigger for bacterial outbreaks and a high percentage of mortality with no identifiable cause.

## 5. Conclusions

Results from this study show that causes of mortality varied within and between sites. For lumpfish in land-based facilities as well as those deployed in small-scale sea pens, the primary cause of mortality was identified as pathogenic, while for lumpfish deployed in large-scale sea pens, transporting, grading, and mechanical delousing were the primary causes of mortality. The results indicate that more research is required to clarify best practices in both commercial hatcheries and salmon cages. Continuous health and welfare monitoring are essential to help identify when and what procedures and operations are detrimental and thus adapt and improve practices.

**Author Contributions:** P.R., A.K.D.I. and L.B. compiled and reviewed findings and analysed new data, wrote the article, and reviewed the manuscript. All authors have read and agreed to the published version of the manuscript.

**Funding:** Financial support was given by the Norwegian Seafood Research Found (EFFEKTIV 901652, DOKUMENTAR 901692) and the Icelandic Research Council (Rannís, 186971-0611).

**Institutional Review Board Statement:** The study was conducted according to the Animal Welfare Act (LOV- 2009-06-19-97). The present field trials were approved by the local responsible laboratory animal science specialist under the surveillance of the Norwegian Animal Research Authority (NARA) and registered by the authority by the following registration numbers: FOTS ID 7475, FOTS ID 20736 and FOTS ID 8246.

**Informed Consent Statement:** Not applicable.

**Data Availability Statement:** Not applicable.

**Acknowledgments:** The authors would like to thank the technical staff at the lumpfish production sites involved for valuable assistance and data sharing.

**Conflicts of Interest:** The authors declare no conflict of interest.

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
