# Peer review of "Causes of Mortality and Loss of Lumpfish Cyclopterus lumpus"

_fishes, doi:10.3390/fishes7060328_

Round 1

Reviewer 1 Report

“The authors conducted very important research to help improve the production quality of the very promising lumpfish with the data and conclusions obtained during the research”.

Author Response

Manuscript number: fishes-2006920

Title: Causes of mortality and loss of lumpfish Cyclopterus lumpus

Comment to authors

“The authors conducted very important research to help improve the production quality of the very promising lumpfish with the data and conclusions obtained during the research”.

Question for authors:

Do they have available data on whether the fish groups examined in the research came from domesticated or wild-caught individuals?

  • Yes, and this is indicated in the third column of Table 1 (origin: W, wild or W/F – 1 generation domesticated individuals).

General recommendation

The obtained results and conclusions will be well utilized in order to increase the production volume of lumpfish, however, it will be advisable to carry out further similar researches in the future so that more data will be available to us. After the recommended corrections, I recommend the article for acceptance.

  • We fully agree and are in the process of publishing more similar data that has been gathered in our studies.

Review

The address (country and/or county) of the company producing the fish feed (and every company in manuscript) must be written in parentheses throughout the manuscript, where such is mentioned.

  • This information has been added throughout the ms.

Line 26: Generally: words from “Title” are not to be repeated in “KW” (mortality, loss, lumpfish). Change it for example: pathogenic, mechanical delousing, transporting etc...

  • We have changed this.

Line 28: In the "Introduction" section, there should be a paragraph about the generally and widely used hatchery practice for lumpfish.

  • We have added a paragraph about the hatchery practice for lumpfish.

Line 69: The statistical method/methods used in the study should also be described in the "Materials and methods" section.

  • We have added a section describing the statistical methods in the M&M section.

Line 79: In the case of "Table 1", an explanation of the abbreviations should be indicated below the table so that the table can be interpreted independently.

  • We have added explanation of the abbreviations.

Table 3 and Table 4: Do not always have decimal value. It is necessary to correct so that all numbers are marked to the same decimal value.

  • All percentage numbers have decimal values. Information about number of fish does not have decimal values.

Figure 7: In the explanation of the figure, it is necessary to explain the abbreviations so that the figure can be interpreted independently.

  • We have added this information in the figure legend.

Reviewer 2 Report

The objective of the work was to evaluate the main causes of mortality in lumpfish, concluding that there were differences in the causes according to the different sampling sites, with pathogens being the main cause of mortality on a small scale, and in large scale cultures, handling operations accumulated most of the observed mortality. 

In general, the work follows scientific standards, although information on the sampling methods used and the protocols employed to discriminate between the causes of mortality is lacking.

As for the results section, the results obtained after evaluation of the "Health Score" introduced in the material and methods are not shown or discussed, so, I recommend including the results or delete this section from the materials and methods.

There are some typing errors that should be corrected throughout the manuscript.

Author Response

Reviewer 2

The objective of the work was to evaluate the main causes of mortality in lumpfish, concluding that there were differences in the causes according to the different sampling sites, with pathogens being the main cause of mortality on a small scale, and in large scale cultures, handling operations accumulated most of the observed mortality. 

In general, the work follows scientific standards, although information on the sampling methods used and the protocols employed to discriminate between the causes of mortality is lacking.

  • We have added the following paragraph to the ms.
  • All lumpfish deployed at Gifas facilities are routinely assessed for health status as mentioned below which is based on external morphological examination. The system allows for further assessment if lumpfish appear distressed and/or mortalities are recorded. Fish are sampled for analysis of internal parameters that may be directly related to the cause of death/distress. These parameters include such assessments as liver colour, ascites, abnormalities to internal organs including signs of pathogenic presence. A range of tissues are sampled and extensively analysed using the appropriate methods which include PCR, histology, bacteriology, and blood sampling.

As for the results section, the results obtained after evaluation of the "Health Score" introduced in the material and methods are not shown or discussed, so, I recommend including the results or delete this section from the materials and methods.

  • It is correct that the Health Score is not explicatively shown in the Results section, but this system is the underlying assessment system used for monitoring the health status of all fish groups in the study. Reporting trends for the Health Score would be far to extensive manoeuvre for this ms but as a part of our routine monitoring, we feel it is necessary to describe this system in the M&M section.

There are some typing errors that should be corrected throughout the manuscript.

  • We have scrutinized the ms and corrected the typing errors found.